# Response of *Lactiplantibacillus plantarum* NMGL2 to Combinational Cold and Acid Stresses during Storage of Fermented Milk as Analyzed by Data-Independent Acquisition Proteomics

**DOI:** 10.3390/foods10071514

**Published:** 2021-06-30

**Authors:** Min Zhang, Mengke Yao, Tiantian Lai, Hua Zhao, Yihui Wang, Zhennai Yang

**Affiliations:** Beijing Advanced Innovation Center for Food Nutrition and Human Health, Beijing Engineering and Technology Research Center of Food Additives, Beijing Technology and Business University, Beijing 100048, China; 1850201008@st.btbu.edu.cn (M.Z.); 1930201080@st.btbu.edu.cn (M.Y.); 1930201070@st.btbu.edu.cn (T.L.); 2030201025@st.btbu.edu.cn (H.Z.); 2030201018@st.btbu.edu.cn (Y.W.)

**Keywords:** *Lactiplantibacillus plantarum* NMGL2, low temperature stress, acid stress, proteomics, metabolic pathway

## Abstract

To understand the mechanism of tolerance of lactic acid bacteria (LAB) during cold storage of fermented milk, 31 LAB strains were isolated from traditional fermented products, and *Lactiplantibacillus plantarum* NMGL2 was identified with good tolerance to both cold and acid stresses. Data-independent acquisition proteomics method was employed to analyze the response of *L**pb. plantarum* NMGL2 to the combinational cold and acid stresses during storage of the fermented milk made with the strain at 4 °C for 21 days. Among the differentially expressed proteins identified, 20 low temperature-resistant proteins and 10 acid-resistant proteins were found. Protein interaction analysis showed that the low temperature-resistant proteins associated with acid-resistant proteins were Hsp1, Hsp2, Hsp3, CspC, MurA1, MurC, MurD, MurE1, and MurI, while the acid-resistant proteins associated with low temperature-resistant proteins were DnaA, DnaK, GrpE, GroEL, and RbfA. The overall metabolic pathways of *L**pb. plantarum* NMGL2 in response to the stresses were determined including increased cell wall component biosynthesis, extracellular production of abundant glycolipids and glycoproteins, increased expression of F_1_F_o_-ATPase, activation of glutamate deacidification system, enhanced expression of proteins and chaperones associated with cell repairing caused by the acidic and cold environment into the correct proteins. The present study for the first time provides further understanding of the proteomic pattern and metabolic changes of *L**pb. plantarum* in response to combinational cold and acid stresses in fermented milk, which facilitates potential application of *L**pb. plantarum* in fermented foods with enhanced survivability.

## 1. Introduction

Functional lactic acid bacteria (LAB) have been widely applied in fermented milk that may be attributed with various functions such as promoting immunity [1], improving intestinal digestion [2], regulating intestinal flora [3], alleviating rheumatoid arthritis [4], relieving major depression [5], and inhibiting gastrointestinal inflammation [6]. During the cold storage of fermented milk, LAB strains are exposed to long-time low temperature and acid stresses, leading to sublethal or even lethal effects on the strains, thus the reduced functionality of the products specially at the later stage of the storage [7,8]. However, the response and the relevant mechanism of the LAB strains against the combined stresses of cold and acid in fermented milk have not been studied.

Cold shock reactions typically occur when exponential growth cultures move from their optimal growth temperature to lower temperatures. In most bacteria, e.g., *B**acillus subtilis*, a temperature downshift causes a transient cell growth arrest, during which general protein synthesis is severely inhibited. However, under these conditions, the synthesis of cold shock proteins (Csps) is triggered. Eventually, the synthesis of these proteins decreases, the cells become acclimated to the low temperature, and growth resumes [9]. Csps have been shown to regulate the expression of cold-induced genes such as anti-terminators [10]. Cold-stress genes such as *cspC*, *cspL*, and *cspP* have been identified in *Lactiplantibacillus plantarum* [11]. Recently, small heat shock (*shs*) genes have been shown to be induced by cold in *Lpb**. plantarum*, and a role for heat shock proteins (Hsps) in preventing damage by low temperature has been suggested [12].

Bacterial acid-stress tolerance involves different mechanisms such as F_1_F_0_-ATPase, glutamate decarboxylase system, arginine deiminase pathway [13], abundance of chaperones or DNA repair system [14], and alterations in the composition of the cell envelope [15]. A series of bacterial anti-acid components have been found, mainly including genes acting as molecular chaperone proteins, regulatory factors, sigma factors, non-coding sRNAs, and transport (membrane) proteins [16]. Moreover, to maintain the equilibrium conditions necessary for cell survival under acid stress, the transport of various substrates including peptides, sugars, amino acids, vitamins and ions is required. The effects of low-pH stress on proteome and transcriptome levels have been determined in multiple LAB strains such as *L. casei* [17], *L. reuteri* [18], *L. rhamnosus* [19], and *Bifidobacterium longum* [20]. Comparison of the protein maps of *B. longum* ssp. *longum* NCIMB 8809 with its acid-resistant mutant indicated that the glutamate deamination pathway played a role in neutralizing internal protons through the production of ammonium [20]. A comparative proteomic study of *Lacticaseibacillus casei* Zhang and its acid-resistant mutant showed that several key proteins associated with cellular metabolism, translation, DNA replication, and chaperones were induced to protect the cells of the mutant [21].

*L**pb.**plantarum* strains are commonly used as the starter for dairy, fermenting vegetables, soy products and meat [22]. These strains have been characterized with anti-oxidation, cholesterol-lowering, and immunity-enhancing functions [23,24]. Although there have been studies on the tolerance of *L**pb.*
*plantarum* against low temperature and acid, the responses of *L**pb. plantarum* at the protein level to combinational cold and acid stresses have not been investigated. In the present study, isolation and identification of an acid and low temperature-resistant LAB strain, namely *L**pb.*
*plantarum* NMGL2, were carried out, and responses of the strain to cold and acid stresses during storage of fermented milk were studied by data-independent acquisition (DIA) proteomics. By using liquid chromatography-tandem mass spectrometry (LC-MS/MS) in the DIA mode, the differentially expressed proteins in *L**pb.*
*plantarum* NMGL2 under different low temperatures and storage time of fermented milk were identified and quantitatively analyzed to gain insight into the mechanism of tolerance of the strain against combinational cold and acid stresses.

## 2. Materials and Methods

### 2.1. Isolation of LAB Strains

Isolation of LAB from traditional naturally fermented products was performed using a previously described method [25] with some modifications. Each of the samples of traditional Inner Mongolia cheese, Gansu and Xinjiang pickle and pickle juice was completely immersed in 100 µL de Man, Rogosa, and Sharpe (MRS) broth and cultured at 37 °C for 24 h. Then, the samples were homogenized and serially diluted with sterile distilled water from 10^−1^ to 10^−8^. 100 µL of different dilutions of each sample was spread on the MRS agar containing 0.3% CaCO_3_ and incubated at 37 °C for 48 h. Single colony with clear zone was picked out. Subsequently, the colonies were streaked on the MRS agar until pure colonies were obtained. The pure strains were morphologically observed by microscope and stored at −80 °C. The isolated LAB strains were initially evaluated for their growth in MRS broth at 37 °C for 18 h, and the viable counts of the strains were determined by plate counting using MRS agar [26].

### 2.2. Identification of LAB Strains

The isolates were identified according to their 16S rRNA gene sequences. The genome DNA was extracted by bacterial genomic DNA extract kit (San-gon, Shanghai, China). PCR amplification of the 16S rRNA gene fragments was performed using primers 27F (5′-AGAGTTTGATCCTGGCTCAG-3′) and 1492R (5′-GGTTACCTTGTTACGACTT-3′), as described earlier [27]. The amplification reaction system had a volume of 25 μL as follows: 12.5 μL 2×Es Taq MasterMix (Dye), 1 μL of each primer, 1 μL template DNA and 9.5 μL DNAse-free water. The program was conducted as follows, pre-degeneration (94 °C, 2 min); 30 cycles: degeneration (94 °C, 30 s), annealing (55 °C, 30 s), extension (72 °C, 30 s); final extension (72 °C, 2 min). The PCR products were separated on 1% agarose gel and visualized under UV light (300 nm) after electrophoresis [28]. At last, amplified products were sequenced by San-gon and sequence Blast of each isolate was completed on the NCBI database.

### 2.3. Tolerance to Low Temperature

MRS broth as prepared and inoculated with 2% overnight bacterial suspensions of all above-mentioned isolates. The inoculated MRS was cultured at 10 °C and 4 °C, respectively. OD_600 nm_ values of each fermentation broth were determined every 24 h from 0 h to 72 h by ultraviolet-visible spectrophotometer UV-1000 (TianMei, Shanghai, China) [29]. The well-grown strains were selected for further in vitro assessment.

### 2.4. Tolerance to Low pH

MRS broth (pH 6.0, 4.0 and 3.0) was prepared and inoculated with 2% overnight bacterial suspensions of all above-mentioned isolates. The strains were cultured in an incubator at 37 °C. OD_600 nm_ values of each fermentation broth were determined every 4 h from 0 to 20 h by UV-1000 (TianMei, Shanghai, China). The well-grown strains were selected for further in vitro assessment.

### 2.5. Scanning Electron Microscopy

The morphological characteristics of *Lpb. plantarum* NMGL2 were determined under acidic and low temperature conditions. Liquid nitrogen was used to freeze the samples, which were then freeze-dried. The dried samples were pasted onto a tape and gold powder was sprayed. The microstructure was observed on s4800 SEM (HITACHI, Tokyo, Japan).

### 2.6. Preparation of Fermented Milk

*Lpb. plantarum* NMGL2 was grown in MRS broth at 37 °C for 24 h, and this procedure was repeated three times. The activated *Lpb. plantarum* NMGL2 suspension was added at 4% with 10^10^ colony forming units (cfu)/mL to the reconstituted skim milk (12% skim milk powder in deionized water). Milk fermentation was performed at 37 °C till pH 4.5, cooled down, and stored at 4 °C for 21 days. Samples were taken at day 1, 7, 14, and 21 for subsequent proteomic studies. Viable counts expressed as log CFU/mL of the fermented milk samples were determined by plate counting on MRS agar at 37 °C for 24 h.

### 2.7. Protein Extraction and Digestion

Lysis buffer (1% SDS, 8 M urea, 1× Protease Inhibitor Cocktail (Roche Ltd. Basel, Switzerland) was added into the fermented milk samples, vibrated and milled for 400 s for three times. The samples were then lysed on ice for 30 min and centrifuged at 12,000× *g* for 15 min at 4 °C. The supernatant was collected and transferred to a new Eppendorf tube.

100 μg of the supernatant per condition in the Eppendorf tube was adjusted to the final volume of 100 μL with 8 M urea. The samples were added with 2 μL of 0.5 M TCEP and incubated at 37 °C for 1 h, then added with 4 μL of 1 M iodoacetamide and incubated at room temperature for 40 min with protection from light. After that, five volumes of −20 °C pre-chilled acetone was added to precipitate the proteins overnight at −20 °C. The precipitates were washed twice by 1 mL pre-chilled 90% acetone aqueous solution and then re-dissolved in 100 μL 100 mM TEAB. Sequence grade modified trypsin (Promega, Madison, WI, USA) was added at the ratio of 1:50 (enzyme: protein, weight: weight) to digest the proteins at 37 °C overnight. The peptide mixture was desalted by C18 ZipTip, quantified by Pierce™ Quantitative Colorimetric Peptide Assay (23275), and lyophilized by SpeedVac. Four group samples were selected, and each group was represented by three biological replicates. For library generation by DDA, all the samples were pooled as a mixture and fractionated by high pH separation. The samples were processed by DIA individually to assess the proteome differences. MS1 and MS2 data were all acquired, and samples acquisition by random order. The iRT kit (Ki3002, Biognosys AG, Basel, Switzerland) was added to all of the samples to calibrate the retention time of extracted peptide peaks.

### 2.8. Liquid Chromatography-Mass Spectrometry Analysis

The peptides were re-dissolved in 30 μL solvent A (A: 0.1% formic acid in water) (Merck KGaA, Darmstadt, Germany) and analyzed by on-line nanospray LC-MS/MS on an Orbitrap Q Exactive HF coupled to EASY-nLC 1200 system (Thermo Fisher Scientific, Waltham, MA, USA). 2 μL peptide sample was loaded onto the analytical column (Acclaim PepMap C18, 75 μm × 25 cm, Merck KGaA, Darmstadt, Germany) with 120-min gradient, from 5% to 30% B (B: 0.1% formic acid in ACN). The column flow rate was maintained at 250 nL/min. The electrospray voltage of 2 kV versus the inlet of the mass spectrometer was used.

The mass spectrometer was run under data independent acquisition mode, and automatically switched between MS and MS/MS mode. The parameters was: (1) MS: scan range (m/z) = 350–1500; resolution = 60,000; AGC target = 3 × 10^6^; maximum injection time = 50 ms; (2) HCD-MS/MS: resolution = 30,000; AGC target = 1 × 10^6^; collision energy = 28; (3) DIA was performed with variable Isolation window, and each window overlapped 1 m/z, and the window number was 42.

### 2.9. Data Analysis for LC-MS

Raw Data of DDA were processed and analyzed by Spectronaut 13 (Biognosys AG, Basel, Switzerland) with default settings to generate an initial target list, which contained 15,199 precursors, 11,564 peptides, 2282 proteins and 2053 protein group. Spectronaut was set up to search the database of Bos_taurus_201907.fasta (ver201907, 23,858 entries) and Lactobacillus_plantarum_201907.fasta (ver201907, 3087 entries) assuming trypsin as the digestion enzyme. Carbamidomethyl (C) was specified as the fixed modification. Oxidation (M) was specified as the variable modifications. Qvalue (FDR) cutoff on precursor and protein level was applied 1%.

Raw Data of DIA were processed and analyzed by Spectronaut 13 (Biognosys AG, Basel, Switzerland) with default settings, Retention time prediction type was set to dynamic iRT. Data extraction was determined by Spectronaut 13 based on the extensive mass calibration. Spectronaut Pulsar 13 will determine the ideal extraction window dynamically depending on iRT calibration and gradient stability. Qvalue (FDR) cutoff on precursor and protein level was applied 1%. Decoy generation was set to mutated which similar to scrambled but will only apply a random number of AA position swamps (min = 2, max = length/2). All selected fregments passing the filters are used for quantification. MS2 interference will remove all interfering fragment ions except for the 3 least interfering ones. The average top 3 filtered peptides which passed the 1% Qvalue cutoff were used to calculate the major group quantities. After Welch’s ANOVA Test, different expressed proteins were filtered if their *p*-value < 0.01 and fold change >2.

### 2.10. Quantitative Data Analysis

Only proteins detected and quantified in all runs (three biological replicates) were included in the data set. To perform a significance test, the Students’ t-test was calculated. After Welch’s ANOVA Test, different expressed proteins were filtered if their *p-*value < 0.01 and fold change >2 or less than 0.5 was defined as being “significantly” regulated in protein quantity.

### 2.11. Bioinformatics Analysis

The differentially expressed proteins were mapped to Gene Ontology (GO) terms (http://www.geneontology.org, 5 Jun 2020) for functional classification and Kyoto Encyclopedia of Genes and Genomes (KEGG) pathways (http://www.kegg.jp/, 5 June 2020) for predicting the main metabolic pathways.

## 3. Results and Discussion

### 3.1. Isolation and Identification of Lactic Acid Bacteria

A total of 31 LAB strains were isolated from traditional fermented products, and among these strains 21 of them showed better growth in MRS broth at 37 °C for 18 h with viable counts greater than 8 log CFU/mL (Appendix A). Indentification by 16S rRNA gene sequencing showed that there were 11 strains of *Lpb. plantarum*, seven strains of *Levilactobacillus brevis*, two strains of *Lacticaseibacillus casei* and one strain of *Lactiplantibacillus pentosus*. These 21 strains with good growth characteristics at 37 °C were selected for further evaluation of their growth under low temperature condition.

### 3.2. Tolerance to Low-Temperature and Low pH

Growth of the selected 21 LAB strains at different low temperatures (10 °C and 4 °C) was determined (Table 1). As expected, the LAB strains showed better tolerance with much better growth at 10 °C than at 4 °C for 72 h, except *L**pb**. plantarum* p2 that was inhibited with little growth at the low temperatures. Among the 21 strains, five of them such as *L**pb. plantarum* NMGL2, *Lcb**. casei* JS2, *L**pb. plantarum* P5, *Lab**. brevis* PC7, and *L**pb. plantarum* NMGL1 tolerated better with relatively better growth at 4 °C for 72 h (ΔA600 ≥ 0.40). These 5 strains were subjected to further evaluation of tolerance to low pH.

The effect of acidic condition (pH 6.0, 4.0, and 3.0) on the viability of the 5 selected LAB strains was further assessed (Table 2). With the decrease of pH, growth of these strains was generally more inhibited as indicated by the lower OD_600 nm_ values at lower pH. Among the 5 strains, *L**pb. plantarum* NMGL2 and *Lcb**. casei* JS2 showed better tolerance to low pH with better growth (ΔA600 ≥ 0.40) at pH 3.0 for 20 h. Considering both the data of low temperature and low pH tolerance, *L**pb.*
*plantarum* NMGL2 was selected for further proteomic study. The 16S rDNA sequence of *L**pb. plantarum* NMGL2 (Appendix A) had 99% homology with that of *L**pb. plantarum* 8M-21 (GenBank: MK049958.1) as indicated in the evolutionary tree of the strain (Figure 1).

### 3.3. Morphological Change of Lpb. plantarum NMGL2 at Low Temperature and Low pH

Morphological change of *L**pb. plantarum* NMGL2 at different temperatures and pH values were studied by SEM. As shown in Figure 2, the bacterial surface of *L**pb. plant**aru**m* NMGL2 appeared to be smooth at 37 °C or at pH 6.0. With the decrease of temperature from 37 °C to 10 °C, or from pH 6.0 to pH 4.0, the bacteria tended to shrink with slightly corrugated surface. Further decrease to 4 °C or pH 3.0 resulted in obviously coarse surface of the bacteria probably due to change of cell wall structure or even cell damage under the cold or acidic condition. Cold stress might compel the bacteria to form a protective layer around the bacterial surface, or express cold shock proteins that could function as cryoprotectants [28]. Acid stress might increase bacterial synthesis of exopolysaccharides or peptidoglycan to strengthen the cell wall structure and decrease the permeability of the cell membrane in order to maintain the normal life activities of the cell [30].

### 3.4. Viability of Lpb. plantarum NMGL2 in Fermented Milk at Low Temperature

*Lpb. plantarum* NMGL2 showed good growth in skim milk as observed by the decreased pH value to 4.50 after growth at 37 °C for 20 h. Figure 3 shows the change of bacterial viable counts in the fermented milk samples of *Lpb. plantarum* NMGL2 throughout the 21 days of storage at 4 °C. In the first 14 days of storage, the viable counts were above 10 log CFU/g. At the end of storage (21 day), the samples had viable counts of 9.92 ± 0.05 log CFU/g, suggesting good survivability of the strain during cold storage in fermented milk.

### 3.5. Proteomic Data Analysis by PCA and PLS-DA

DIA-based proteomics study was performed to analyze the response of *L**pb. plantarum* NMGL2 to the combinational cold and acid stresses during storage of the fermented milk made with the strain at 4 °C for 21 days. Principal component analysis (PCA) was carried out separately on each data set using the R function prcomp from the stats package with default parameters. PLS discriminant analysis (PLS-DA) was used for classification and discrimination of problems on the basis of a classical PLS regression (with a regression mode) by using mixOmics package (https://CRAN.R-project.org/package=mixOmics, 11 May 2020).

PCA and PLS-DA methods were used to analyze the differentially expressed proteins in *L**pb.*
*plantarum* NMGL2 in fermented milk stored at low temperature (4 °C) for days 1, 7, 14, and 21 (Figure 4). The results showed day 1 and day 7 components on the right side of the score, and day 14 and day 21 components on the left side of the score. The PLS-DA score graph indicated that the four groups of samples were distributed in four different regions of the score graph. The day 1 and day 7 groups could be classified into the same group with significant differences between groups, while the day 14 and day 21 groups could be classified into the same group with significant differences between groups.

### 3.6. Differentially Expressed Proteins

Proteomic analysis on *L**pb. plantarum* NMGL2 fermented milk with different storage time showed that a total of 851 proteins were detected on the first day. The total protein number on day 7 was 771, and the total protein numbers on days 14 and 21 were 1661 and 1683 respectively. On day 7, compared with the day 1, there were 115 differentially expressed proteins, among which 25 proteins were up-regulated and 90 proteins down-regulated. On day 14, there were 932 differentially expressed proteins, among which 930 proteins were up-regulated and two proteins down-regulated relative to day 1. On day 21, there were 1032 differentially expressed proteins, among which 1030 proteins were up-regulated and two proteins down-regulated relative to day 1 (Figure 5A).

The differentially expressed proteins were enriched by the KEGG pathway to five classifications: metabolism, human diseases, genetic information processing, environmental information processing, and cellular processes. The metabolic pathways were mainly xenobiotic biodegradation and metabolism, nucleotide metabolism, metabolism of terpenoids and polyketides, metabolism of other amino acids, metabolism of cofactors and vitamins, lipid metabolism, global and overview maps, energy metabolism, carbohydrate metabolism, biosynthesis of other secondary metabolites, and amino acid metabolism. One of the enrichment pathways in human diseases was antimicrobial drug resistance. The pathways that were mainly enriched in genetic information processing were translation, transcription, replication, repair, and folding, as well as sorting and degradation. Two of the pathways that were enriched in environmental information processing were signal transduction and membrane transport. One pathway of enrichment in cellular processes was cellular community prokaryotes (Figure 5B).

The differentially expressed proteins were enriched through the GO pathway, and in biological process. They were mainly enriched for oxidation-reduction process, phosphorylation, translation, proteolysis, regulation of transcription, DNA-templating, transmembrane transport, carbohydrate metabolic process, cell division, phosphoenolpyruvate-dependent sugar phosphotransferase activity, and nucleic acid phosphodiester bond hydrolysis. In cellular components, it was mainly enriched in the cytoplasm, integral component of membranes, plasma membrane, ribosome, ATP-binding cassette (ABC) transporter complex, cell, cell surface, chromosome, membrane, proton-transporting ATP synthase complex, and catalytic core F(1). In molecular function, it was mainly enriched for ATP binding, DNA binding, metal ion binding, hydrolase activity, ATPase activity, structural constituents of ribosome, zinc ion binding, magnesium ion binding, DNA-binding transcription factor activity, and rRNA binding (Figure 5C).

### 3.7. Screening of Low Temperature and Acid-Resistant Proteins

The proteins related to acid resistance and low temperature resistance were searched from the differentially expressed proteins. The screened proteins were further analyzed for interaction with each other through String software to find the associated proteins. A total of 20 low temperature-resistant proteins (Hsp1, Hsp2, Hsp3, MurA1, MurB, MurC, MurD, MurE1, MurF, MurI, DacA, DacA1, DacB, CspC, GaLE1, GaLE2, GaLK, GaLM1, GaLM3, and GaLU), and 10 acid-resistant proteins (DnaA, DnaC, DnaI, DnaJ, DnaK, DnaN, DnaX, GrpE, GroEL, and RbfA) were obtained (Figure 6A). The 30 genes were detected at significantly higher levels on days 14 and 21 than on days 1 and 7. Some acid-resistant proteins (DnaA, DnaK, GrpE, GroEL, and RbfA) and low temperature resistant proteins (Hsp1, Hsp2, Hsp3, CspC, MurA1, MurC, MurD, MurE1, and MurI) were correlated with each other as shown in the protein interaction network diagram (Figure 6B).

### 3.8. Parallel Reaction-Monitoring (PRM) Validation

To verify the abundance of proteins in the DIA (data-independent acquisition) analysis, PRM was performed to confirm 30 proteins of interest, the abundance levels of which were detectable in our study: DnaA, DnaC, DnaI, DnaJ, DnaK, DnaN, DnaX, GrpE, GroEL, RbfA, Hsp1, Hsp2, Hsp3, MurA1, MurB, MurC, MurD, MurE1, MurF, MurI, DacA, DacA1, DacB, CspC, GaLE1, GaLE2, GaLK, GaLM1, GaLM3, and GaLU. The results (Figure 7A,B) of the PRM showed that all confirmed proteins were significantly enriched in association with cold and acid stresses. These protein trends are consistent with findings from DIA data and thus support the reliability of DIA measurements (Appendix A).

### 3.9. Metabolic Pathway Analysis

Lactic acid bacteria are widely used in food fermentation industry as lactic starters, and tolerance of the LAB strains to cold and acid conditions is of importance for their survivability during cold storage of the fermented products. Based on the proteomic data described above, the overall metabolic pathways involved in response of *L**pb. plantarum* NMGL2 to combinational cold and acid stresses during storage of fermented milk is shown in Figure 8. Under the combinational cold and acid stresses, biosynthesis of cell wall components increased in *L**pb. plantarum* NMGL2 involving expression of several proteins such as DanA, DanJ, DanK, DnaN, DnaX and GuaC. Meanwhile, expression of several other proteins such as GalE1, GalE2, GalK, GalM1, GalM3, and GalU promoted the synthesis of extracellular glycolipids and glycoproteins to provide protection arround the bacteria cells. Furthermore, proton pump (F_1_F_o_-ATPase) was activated involving expression of Dnak and GroEL to maintain normal physiological pH values in the cytoplasm. Expression of MurC, MurD, and MurI participated in the decarboxylation of glutamate system to produce an alkaline substance to maintain the integrity of the cell. Molecular chaperones such as Cspc, DnaK, GrpE, Hsp1, Hsp2, and hsp3 participated in various cell repairing related activities upon stress-induced cell damage such as DNA repair and recombination, protein folding, protein renaturation, protein protection from denaturation, amino acid biosynthesis and metabolism. More detailed analysis of the main metabolic pathways involved in the stress response of *L**pb. plantarum* NMGL2 was discussed below.

Enhancement of cell wall component biosynthesis acted as a self-protective mechanism and the first line of defense of *L**pb. plantarum* NMGL2 against environment stresses. Some proteins were shown to be up-regulated to increase total peptidoglycan production, such as DacA1, DacB, MurA1, MurB, MurC, MurD, MurE1, and MurF. Meanwhile, production of glycolipids and glycoproteins increased in *L**pb. plantarum* NMGL2 involving several differentially expressed proteins such as GalE1, GalE2, GalK, GalM1, GalM3, and GalU. Glycolipids and glycoproteins are structural components of the extracellular matrix that support, connect, and buffer for bacterial protection, and they play a role in cell recognition, adhesion, and migration, as well as regulation of cell proliferation and differentiation [12]. Previously, *S**taphylococcus aureus* was also shown to increase the cell wall peptidoglycan biosynthesis after prolonged cold exposure [31]. It was reported that higher proton (H^+^) concentration in an acidic environment and a soluble undissociated acidic substance (e.g., lactic acid) could enter the cell to acidify the cytoplasm, resulting in cellular metabolism disorder and even cell death [21]. Probiotic lactic acid bacteria developed various mechanisms to resist the adverse acidic environment [32]. Proton pump (F_1_F_o_-ATPase) in lactic acid bacteria played a crucial role in maintaining the stability of pH values in cells at low pH values, and the main proteins involved in this process were Dnak, GroEL, Htr, Clp ATPases, and Lo18 [20]. Some cationic transporter ATPases (such as K^+^-ATPas) were also reported to play a role in maintaining the intracellular stability of pH values through the exchange of H^+^ and K^+^ [32,33]. In *L**pb. plantarum* NMGL2, two differentially expressed proteins of Dnak and GroEL were found to be associated with the acid resistant mechanism of the strain. Furthermore, under the acidic condition the glutamate deacidification system converted glutamic acid in the cell into γ-aminobutyric acid, resulting in depletion of hydrogen ions and increase of the intracellular pH, while the excretion of γ-aminobutyric acid as an alkaline substance increased the extracellular pH [34]. A number of gene-expressed proteins such as MurC, MurD, and MurI, were found in *L**pb.*
*plantarum* NMGL2 to be involved in the decarboxylation of glutamate to produce an alkaline substance that maintained the integrity of the cell.

Some molecular chaperones play an important role in many stress-related cell protection and repairing properties, such as DNA repair and recombination, protein folding, protein renaturation, and protein protection from denaturation [35]. Fermented milk during storage at 4 °C resulted in high expression of the molecular chaperone proteins (DnaK, GrpE, Hsp1, Hsp2, and Hsp3) in *L**pb. plantarum* NMGL2, which were involved in cell repairing, e.g., protein refolding of the misfolded proteins caused by the environment stresses. In *L**pb. plantarum* L67 isolated from infant feces, expression of cold shock protein and ATPase-active substances was stimulated at 5 °C for 6 h [7], while *L.*
*paracasei* and *L. mali* isolated from wine were found to increase expression of antifreeze and ice-nucleation proteins when stimulated at −20, −80, and −196 °C for 1 h [36]. In *L**actococcus lactis*, synthesis of some molecular chaperones (such as GroEL, GroES, GnaK, ClpE, and GrpE) could be induced at pH 4.5 [8]. Acidic conditions could also induce high expression of some Hsp in LAB strains (such as *Lcb**. casei*) [17]. Furthermore, acid stress resulted in high expression of the molecular chaperone protein (DnaK) in *S**treptococcus mutans* at both transcriptional and protein levels [30]. The Csp (cold shock protein) family such as CspA, CspB and CspC, as well as Hsp (heat shock protein) family such as Hsp1, Hsp2 and Hsp3, could respond to cold shock and acted as RNA chaperons to improve the protein synthesis capacity [37,38]. In *L**pb. plantarum* NMGL2, several differentially expressed proteins such as DanA, DanJ, K, DanN, DanX, and GuaC, were found to involve in amino acid biosynthesis and metabolism in response to the stresses. In addition to supplying materials for protein synthesis, amino acids, e.g., histidine and lysine, have various biological functions such as being a precursor of physiologically activate substrates, regulating hormone secretion and cell turnover [39].

## 4. Conclusions

*L**pb. plantarum* NMGL2 isolated from traditional fermented products showed good tolerance to low temperature (4 °C) and acid (pH 3.0). When *L**pb. plantarum* NMGL2 was used to prepare fermented milk, proteomic analysis of the strain in response to combinational cold and acid stresses during the storage for 1, 7, 14, and 21 days at 4 °C revealed 20 low temperature-resistant proteins and 10 acid-resistant proteins. Further protein interaction analysis showed that the low temperature-resistant proteins associated with acid-resistant proteins were Hsp1, Hsp2, Hsp3, CspC, MurA1, MurC, MurD, MurE1, and MurI, while the acid-resistant proteins associated with low temperature-resistant proteins were DnaA, DnaK, GrpE, GroEL, and RbfA. The metabolic pathways involved in response of *L**pb. plantarum* NMGL2 to the stresses included increased cell wall component biosynthesis, extracellular production of abundant glycolipids and glycoproteins, increased expression of F_1_F_o_-ATPase, activation of glutamate deacidification system, enhanced expression of proteins and chaperones associated with cell repairing. Future study would focus on identification of key genes and proteins related to the viability of *L**pb. plantarum* NMGL2 under combinational cold and acid stresses for its potential application in functional foods with improved survivability during storage.

## Figures and Tables

**Figure 1 foods-10-01514-f001:**
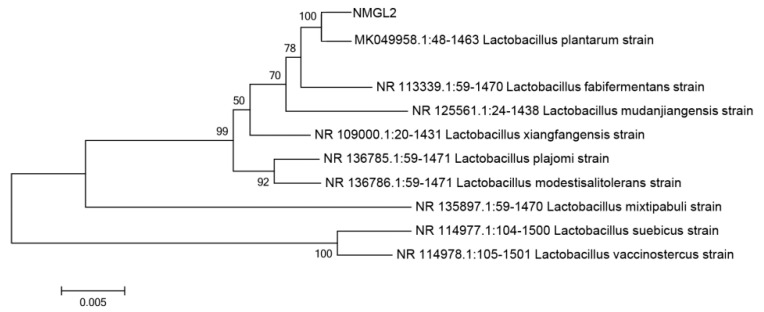
Evolutionary tree of *L**pb. plantarum* NMGL2.

**Figure 2 foods-10-01514-f002:**
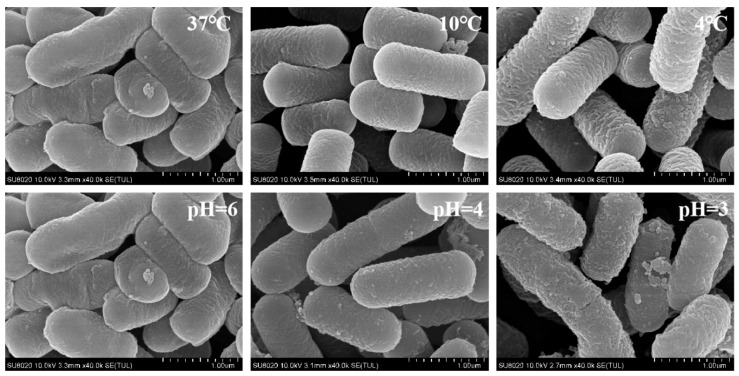
SEM images of *L**pb. plantarum* NMGL2 at different temperatures: 37 °C, 10 °C, and 4 °C and at pH 6.0, 4.0, and 3.0. The magnification was 1.00 µm.

**Figure 3 foods-10-01514-f003:**
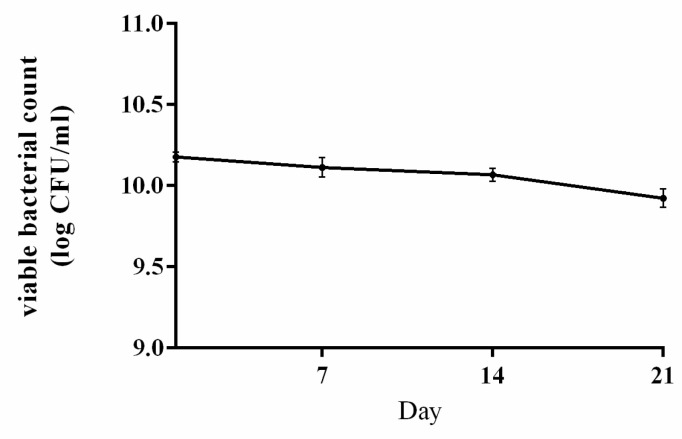
Change in viable count of fermented milk made with *Lpb. plantarum* NMGL2 during 21 days of storage at 4 °C.

**Figure 4 foods-10-01514-f004:**
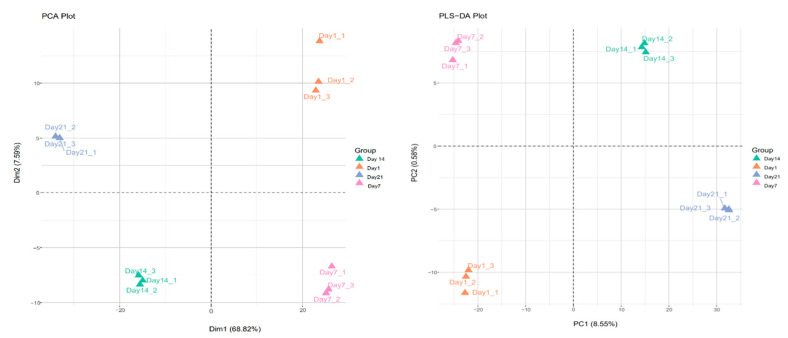
PCA and PLS-DA analyses on the differentially expressed proteins in *L**pb. plantarum* NMGL2 in fermented milk stored at low temperature (4 °C) for 1, 7, 14, and 21 days. (Dim1: Principal component 1. Dim2: Orthogonal component 1. PC1: Measurement of fit. PC2: Measurement of prediction).

**Figure 5 foods-10-01514-f005:**
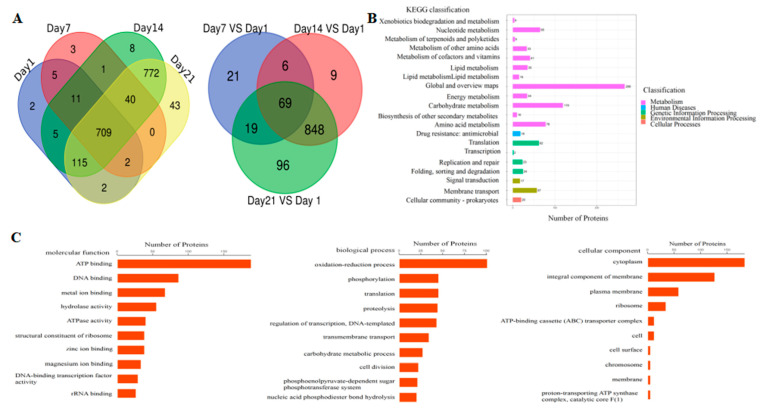
(**A**) The number of protein groups and differentially expressed proteins identified in *L**pb.*
*plantarum* NMGL2 in fermented milk stored at low temperature (4 °C) for days 1, 7, 14, and 21. (**B**) KEGG pathway enrichment result of differentially expressed proteins. (**C**) GO terms enrichment result of differentially expressed proteins.

**Figure 6 foods-10-01514-f006:**
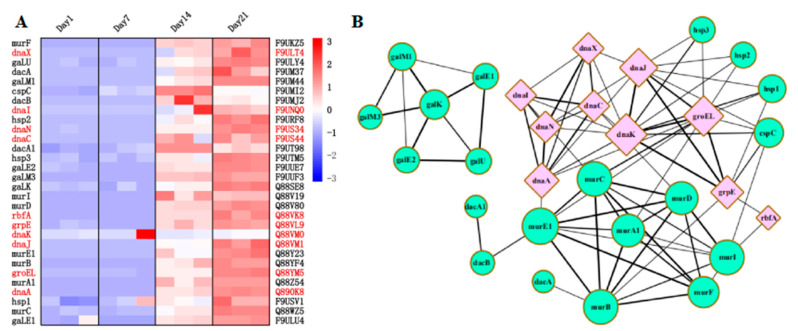
Quantitative analysis of differentially expressed acid-resistant and low-temperature-resistant proteins in *L**pb. plantarum* NMGL2 in fermented milk stored at low temperature (4 °C) for days 1, 7, 14, and 21 (**A**), using String software to visualize the different proteins on the network (**B**).

**Figure 7 foods-10-01514-f007:**
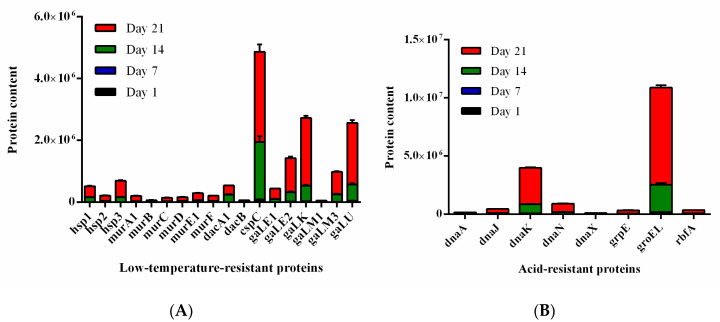
Expression of low-temperature-resistant proteins (**A**) and acid-resistant proteins (**B**) during cold storage of *L**pb. plantarum* NMGL2 fermented milk at days 1, 7, 14, and 21.

**Figure 8 foods-10-01514-f008:**
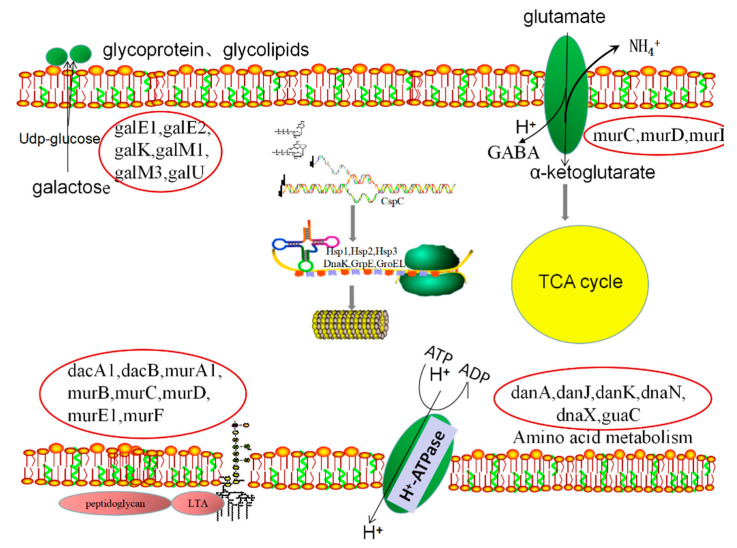
Proteomic pattern of *L**pb. plantarum* NMGL2 in response to combinational cold and acid stresses during cold storage of fermented milk.

**Table 1 foods-10-01514-t001:** Growth of 21 strains of lactic acid bacteria at 10 °C and 4 °C for 72 h (*n* = 3, *x* ± SD).

	OD600 _nm_ Value						OD600 _nm_ Value						
Incubation Time	24 h		48 h		72 h		Incubation Time	24 h		48 h		72 h	
Incubation Temperature	4 °C	10 °C	4 °C	10 °C	4 °C	10 °C	Incubation Temperature	4 °C	10 °C	4 °C	10 °C	4 °C	10 °C
*Lpb. plantarum* NMGL2	0.12 ± 0.02	0.20 ± 0.02	0.25 ± 0.03	0.90 ± 0.04	0.68 ± 0.04	1.11 ± 0.03	*Lpb. plantarum* P4	0.12 ± 0.02	0.15 ± 0.03	0.22 ± 0.03	0.72 ± 0.04	0.33 ± 0.02	0.99 ± 0.04
*Lcb. casei* JS2	0.12 ± 0.03	0.17 ± 0.01	0.22 ± 0.02	0.65 ± 0.03	0.46 ± 0.03	0.86 ± 0.03	*Lpb. plantarum* PC10	0.12 ± 0.03	0.16 ± 0.04	0.18 ± 0.02	0.83 ± 0.03	0.31 ± 0.04	1.08 ± 0.03
*Lpb. plantarum* P5	0.12 ± 0.02	0.15 ± 0.02	0.22 ± 0.03	0.80 ± 0.04	0.45 ± 0.03	1.04 ± 0.02	*Lpb. plantarum* P7	0.12 ± 0.02	0.16 ± 0.04	0.21 ± 0.04	0.33 ± 0.02	0.27 ± 0.01	0.45 ± 0.02
*Lab. brevis* PC7	0.10 ± 0.02	0.13 ± 0.03	0.21 ± 0.01	0.64 ± 0.04	0.45 ± 0.02	0.87 ± 0.04	*Lab. brevis* PC2	0.10 ± 0.02	0.13 ± 0.04	0.21 ± 0.04	0.41 ± 0.03	0.27 ± 0.03	0.59 ± 0.04
*Lpb. plantarum* NMGL1	0.13 ± 0.04	0.16 ± 0.02	0.22 ± 0.02	0.37 ± 0.04	0.41 ± 0.02	0.91 ± 0.03	*Lab. brevis* PC6	0.12 ± 0.01	0.10 ± 0.03	0.21 ± 0.02	0.37 ± 0.02	0.27 ± 0.03	0.48 ± 0.04
*Lcb. casei* JS1	0.12 ± 0.02	0.16 ± 0.01	0.21 ± 0.04	0.66 ± 0.02	0.37 ± 0.02	0.84 ± 0.02	*Lpb. plantarum* P3	0.11 ± 0.02	0.11 ± 0.03	0.20 ± 0.03	0.51 ± 0.04	0.23 ± 0.04	0.79 ± 0.03
*Lab. brevis* PC5	0.11 ± 0.01	0.13 ± 0.04	0.21 ± 0.04	0.66 ± 0.02	0.36 ± 0.02	0.93 ± 0.03	*Lab. brevis* PC3	0.10 ± 0.01	0.13 ± 0.02	0.20 ± 0.02	0.37 ± 0.04	0.22 ± 0.03	0.51 ± 0.03
*Lpb. pentosus* PC11	0.12 ± 0.04	0.17 ± 0.03	0.23 ± 0.04	0.77 ± 0.01	0.35 ± 0.01	0.99 ± 0.02	*Lab. brevis* PC8	0.10 ± 0.01	0.11 ± 0.04	0.20 ± 0.04	0.27 ± 0.03	0.19 ± 0.04	0.40 ± 0.04
*Lpb. plantarum* P6	0.12 ± 0.01	0.15 ± 0.03	0.22 ± 0.02	0.73 ± 0.03	0.35 ± 0.01	0.94 ± 0.01	*Lab. brevis* PC9	0.10 ± 0.03	0.13 ± 0.02	0.20 ± 0.01	0.37 ± 0.02	0.17 ± 0.02	0.51 ± 0.02
*Lpb. plantarum* P1	0.12 ± 0.02	0.15 ± 0.01	0.22 ± 0.03	0.67 ± 0.03	0.34 ± 0.02	0.88 ± 0.01	*Lpb. plantarum* P2	0.12 ± 0.01	0.13 ± 0.01	0.12 ± 0.02	0.14 ± 0.04	0.13 ± 0.02	0.14 ± 0.01
*Lpb. plantarum* NMGL3	0.14 ± 0.01	0.17 ± 0.02	0.25 ± 0.01	0.70 ± 0.04	0.34 ± 0.02	0.97 ± 0.02							

**Table 2 foods-10-01514-t002:** Growth of 5 strains of lactic acid bacteria at pH 6.0, 4.0, and 3.0 (*n* = 3, *x* ± SD).

	OD600 _nm_ Value											
pH	6			4				3				
Incubation temperatures (h)	4	8	12	4	8	12	16	4	8	12	16	20
*Lpb. plantarum* NMGL2	0.86 ± 0.02	1.29 ± 0.01	1.44 ± 0.02	0.35 ± 0.01	0.67 ± 0.02	0.92 ± 0.01	1.22 ± 0.02	0.19 ± 0.01	0.25 ± 0.02	0.31 ± 0.02	0.34 ± 0.02	0.47 ± 0.02
*Lcb. casei* JS2	0.74 ± 0.01	1.21 ± 0.02	1.39 ± 0.02	0.33 ± 0.02	0.62 ± 0.01	0.88 ± 0.02	1.24 ± 0.03	0.17 ± 0.02	0.18 ± 0.01	0.21 ± 0.01	0.45 ± 0.01	0.48 ± 0.03
*Lpb. plantarum* P5	0.74 ± 0.02	1.15 ± 0.02	1.40 ± 0.01	0.31 ± 0.02	0.56 ± 0.01	0.76 ± 0.02	1.07 ± 0.02	0.16 ± 0.01	0.18 ± 0.02	0.19 ± 0.02	0.25 ± 0.01	0.27 ± 0.02
*Lab. brevis* PC7	0.25 ± 0.03	0.50 ± 0.01	0.82 ± 0.02	0.12 ± 0.03	0.19 ± 0.02	0.30 ± 0.01	0.40 ± 0.01	0.10 ± 0.01	0.10 ± 0.02	0.10 ± 0.01	0.12 ± 0.02	0.11 ± 0.03
*Lpb. plantarum* NMGL1	0.88 ± 0.02	1.33 ± 0.02	1.50 ± 0.03	0.41 ± 0.02	0.74 ± 0.02	0.94 ± 0.02	1.28 ± 0.02	0.18 ± 0.02	0.21 ± 0.01	0.25 ± 0.02	0.34 ± 0.01	0.38 ± 0.02

## Data Availability

Not applicable.

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
