# Peer review of "Response of Lactiplantibacillus plantarum NMGL2 to Combinational Cold and Acid Stresses during Storage of Fermented Milk as Analyzed by Data-Independent Acquisition Proteomics"

_foods, 2021, doi:10.3390/foods10071514_

Round 1
Reviewer 1 Report
General comment:
The manuscript by Zhang et al. reports about the response of Lactobacillus plantarum NMGL2 after milk fermentation to acid stresses during cold storage. Paper is prepared according to author instructions. Some of methods are not properly chosen and some of the results are not clearly presented. According to the presented results, low temperature-resistant proteins and acid-resistant proteins were determined, what is the main scientific contribution of the manuscript. However, a lot of improvements are needed in the terms of clarity of the manuscript.
Major concern:
The data in table 1, where growth of the examined LAB strains is expresed as log cfu/ml, do not correspond with data in tables 2 and 3, were growth is expressed as OD600nm value. It is not possible to missure the growth as OD600nm in milk because it is not transparent liquid and the results are not relevant. Therefore, the experimental data in tables 2 and 3 must be expressed as log cfu/ml, and the method for determination of viable cell count must be described in the chapter Material and methods. Otherwise, tables 2 and 3 must be excluded from the manuscript.
Furthermore, there are no scientific contribution regarding results presented in Table 1 were optimal growth conditions were applied. Therefore, table 1 does not contribute to the research results regarding aim of study and must be excluded from the manuscript.
My recomendation is that results presented in tables 1, 2 and 3 can be eliminated, and only results regarding strain Lactobacillus plantarum NMGL2 are relevant for the aim of study described in the manuscript.
According to presented experimental results, the proteomic pattern of cold and acid resistance of L. plantarum strain NMGL2 was studied, rather than „molecular mechanism involved in the tolerance of the strain against combinational cold and acid stresses“, as it is mentioned in the abstract and in tekst of manuscript.
Specific comments:
In the abstract and in the text of manuscript:
The marks of proteins must be written with capital first letters.
The latin names of bacterial species must be written in italic style!
In the Material and Methods:
line 110: In the chapter 2.4. „Tolerance to low pH“ – the incubation temperature is missing!
lines 112-113: The sentence does not make sense „OD600 nm values of each fermentation 112 broth were determined every 4 h from 0 h to 20 h by pH Meter PHS-3E (LeiCi, China).“
In the Results and Discussion:
Lines 221-223: „The 16S rDNA sequence of L. plantarum NMGL2 had 99% homology with that of L. plantarum 8M-21 (Gen-222 Bank: MK049958.1) as indicated in the evolutionary tree of the strain (Figure 1).“ On Figure 1 can be seen 3 different strains with 99% homology with L. plantarum NMGL2, but not the strain L. plantarum 8M-21 as it is mentioned in the sentence of the manuscript.
Figure 6.
Instead of acid-resistant proteins, there are two times presented low-temperature-resistant proteins. Acid resistant proteins presentation is missing.
Figure 7:
There are shown proposed proteomic patern of the strain L. plantarum NMGL2 regarding cold and acid stress response in fermented milk rather than „Overall metabolic pathways in L. plantarum NMGL2 in response to combinational cold and acid stresses during storage of fermented milk“
References:
There are no cited references of the authors of this manuscript and if there are any, regarding the aim of this study, it will be useful to cite it.
There are no cited references from the year 2020-2021. The results of the latest references related to the research area of this manuscript should certainly be discussed.
Author Response
General comment:
The manuscript by Zhang et al. reports about the response of Lactobacillus plantarum NMGL2 after milk fermentation to acid stresses during cold storage. Paper is prepared according to author instructions. Some of methods are not properly chosen and some of the results are not clearly presented. According to the presented results, low temperature-resistant proteins and acid-resistant proteins were determined, what is the main scientific contribution of the manuscript. However, a lot of improvements are needed in the terms of clarity of the manuscript.
Response: Thanks for the comments. More corrections have been made to clarify relevant methods (e.g. identification of LAB, tolerance to low pH) and results (e.g. isolation of LAB) in the revised manuscript. The present study added new data about the low temperature-resistant proteins associated with acid-resistant proteins in Lactiplantibacillus plantarum, and vise versa, and metabolic pathways in response to combinational cold and acid stresses in fermented milk were proposed for the first time.
Major concern:
Comment: The data in table 1, where growth of the examined LAB strains is expresed as log cfu/ml, do not correspond with data in tables 2 and 3, were growth is expressed as OD600nm value. It is not possible to missure the growth as OD600nm in milk because it is not transparent liquid and the results are not relevant. Therefore, the experimental data in tables 2 and 3 must be expressed as log cfu/ml, and the method for determination of viable cell count must be described in the chapter Material and methods. Otherwise, tables 2 and 3 must be excluded from the manuscript.
Response: Thanks a lot for the comments. The data in Table 2 and Table 3 were obtained by
measuring OD600nm in MRS medium, not in milk, to compare the cold and acid tolerance
between strains, and the method used as described ealier was cited with a reference
(determine the bacterial growth by measuring OD600nm) (line 111, 496) . A reference was added about the method for determination of viable cell count (line 93, 488)
Comment: Furthermore, there are no scientific contribution regarding results presented in Table 1 were optimal growth conditions were applied. Therefore, table 1 does not contribute to the research results regarding aim of study and must be excluded from the manuscript.
Response: Thanks for your advice. Isolation of the LAB strains was done following the previously described method using the optimal growth conditions (37℃) to evaluate the growth of the strains, and those (21 strains) showing good growth were considered as generally good strains that were selected for further study. In the revised manuscript, 10 strains showing less growth (not seleted for further study) were added in Tables 1.
Comment: My recomendation is that results presented in tables 1, 2 and 3 can be eliminated, and only results regarding strain Lactobacillus plantarum NMGL2 are relevant for the aim of study described in the manuscript.
Response: Thanks for your advice. To find a strain with good tolerance to low temperature and low pH, 21 LAB strains showing good growth in MRS were initially assayed for tolerance to low temperatue (results shown in Table 2) , and 5 strains showing good tolerance to low temperature were further assayed for tolerance to low pH (results shown in Table 3), and finally Lactiplantibacillus plantarum NMGL2 was selected as the best strain with best tolerance to both low temperature and low pH, which was subject to proteomic study.
Comment: According to presented experimental results, the proteomic pattern of cold and acid resistance of L. plantarum strain NMGL2 was studied, rather than „molecular mechanism involved in the tolerance of the strain against combinational cold and acid stresses“, as it is mentioned in the abstract and in tekst of manuscript.
Response: Thanks for your advice. We have modified the text to be the proteomic pattern and metabolic changes in the revised manuscript.
Specific comments:
Comment: In the abstract and in the text of manuscript:
The marks of proteins must be written with capital first letters.
Response: We have corrected all.
Comment: The latin names of bacterial species must be written in italic style!
Response: We have modified all.
Comment: In the Material and Methods:
line 110: In the chapter 2.4. „Tolerance to low pH“ – the incubation temperature is missing!
Response: We have added the temperature in this part.
Comment: lines 112-113: The sentence does not make sense „OD600 nm values of each fermentation 112 broth were determined every 4 h from 0 h to 20 h by pH Meter PHS-3E (LeiCi, China).“
Response: We have revised this sentence.It is changed to“OD600 nm values of each fermentation broth were determined every 4 h from 0 h to 20 h by UV-1000 (TianMei, China)”.
Comment: In the Results and Discussion:
Lines 221-223: „The 16S rDNA sequence of L. plantarum NMGL2 had 99% homology with that of L. plantarum 8M-21 (Gen-222 Bank: MK049958.1) as indicated in the evolutionary tree of the strain (Figure 1).“ On Figure 1 can be seen 3 different strains with 99% homology with L. plantarum NMGL2, but not the strain L. plantarum 8M-21 as it is mentioned in the sentence of the manuscript.
Response: Thanks for your advice. We redid the comparative analysis and presented the latest comparative results in the manuscript.
Comment: Figure 6.
Instead of acid-resistant proteins, there are two times presented low-temperature-resistant proteins. Acid resistant proteins presentation is missing.
Response: In the manuscript, we have revised figure 6.
Comment: Figure 7:
There are shown proposed proteomic patern of the strain L. plantarum NMGL2 regarding cold and acid stress response in fermented milk rather than „Overall metabolic pathways in L. plantarum NMGL2 in response to combinational cold and acid stresses during storage of fermented milk“
Response: We have modified the description in Figure 7 as “Proteomic pattern of Lpb. plantarum NMGL2 in response to combinational cold and acid stresses during cold storage of fermented milk.”
Comment: References:
There are no cited references of the authors of this manuscript and if there are any, regarding the aim of this study, it will be useful to cite it.
Response: Because this strain of bacteria is newly isolated and purified, and no articles on this strain have been published, but the characteristics of this strain of bacteria have been applied for a patent, and the authorization number is: CN201910916373.6
Comment: There are no cited references from the year 2020-2021. The results of the latest references related to the research area of this manuscript should certainly be discussed.
Response: Thanks for your advice. We have added discussion and analysis of the latest articles to the manuscript.
Reviewer 2 Report
- First of all the authors should revise species names. Lactobacillus plantarum is no more indicated in this way since 2020 (Zheng et al., Int. J. Syst. Evol. Microbiol. 2020;70:2782–2858). Its current name is Lactiplantibacillus plantarum. Actually, two subspecies are recognized: Lactiplantibacillus plantarum subsp. plantarum and Lactiplantibacillus plantarum subsp. argentoratensis. Since your work is specifically aimed to evaluate a stress response, please clarify this point, a given species (even strain) could behave singly, so it has to be clear if the behaviour is species specific or not.
- In general, the abstract should be more descriptive, reporting the main numerical data (percentages, levels, concentrations etc.). As is, the abstract does not stand alone.
- I do not agree with the conclusion “The present study for the first time provides further understanding of the molecular mechanism involved in response of L. plantarum to combinational cold and acid stresses in fermented milk, which facilitates potential application of plantarum in fermented foods with enhanced functionality.” Because no speculations on data are provided. In which way your findings are useful? How could it be of help? I am thinking of cold shock response. Fermentation is carried out at mesophilic temperature, so which is the implication of resistance to cold shock?
- L94-96. If the primers were designed in this work, please provide details on the corresponded positions on a given type strain used for designation. Otherwise, if primers description was published elsewhere, please add reference.
- The strong limit of this work is a control production. Only one strain, the one the authors selected based on in vitro screening, but no comparison with a strain showing the worst characteristic was provided. How can the authors state that the strain chosen behave better than other LAB without a comparison trial? It is as if a control is missing
- It is not clear the criterion followed to identify bacteria. To do so, bacteria should be isolated following a specific objective, which is not clear from the text, since a main hypothesis is not provided in the abstract. If bacteria resistant to acidic stresses should be collected, then it was better to screen a huge number of bacteria for this character and only those showing interesting characteristic be identified.
- The isolation plan itself is not clear. No number of samples, repetition, characteristics of the samples chosen were provided. Why isolate bacteria from pickle serum to produce a fermented milk? It apparently does not make any sense. Please clarify this very crucial point.
- Table 1. This table is not clear too. “Growth of 21 strains of lactic acid bacteria isolated from traditional fermented products in 200 MRS broth at 37°C for 18 h as shown with viable bacterial counts (n = 3, x ± SD).”. In my opinion, the authors should show the levels of viable bacteria in the fermented products used for isolation. This would provide important indications on the possible success of the selected strains in fermented milk. This work does not provide indication on the levels reached in a real food system. Showing the levels reached in synthetic media is not useful, since each bacterium grows alone and is not subjected to the competition with other bacteria. Of course all bacteria grow well in MRS.
- L202-203. The results “These strains showed good growth in MRS broth at 37°C for 18 h, with viable counts 202 from 8 log cfu/mL to about 10 log cfu/mL.” do not provide any useful indication. To be eliminated. It would make more sense to show the growth in milk.
- Levilactobacillus brevis, not Lactobacillus brevis (Zheng et al., 2020)
- Lacticaseibacillus casei (Zheng et al., 2020)
- Lactiplantibacillus pentosus (Zheng et al., 2020)
- please revise species name also in Introduction.
- Figure 1. Evolutionary tree of L. plantarum NMGL2. First of all it should be reported on which gene. Secondly, I am not sure of its significance to support strain selection.
Author Response
- Comment:First of all the authors should revise species names. Lactobacillus plantarum is no more indicated in this way since 2020 (Zheng et al., Int. J. Syst. Evol. Microbiol. 2020;70:2782–2858). Its current name is Lactiplantibacillus plantarum. Actually, two subspecies are recognized: Lactiplantibacillus plantarum subsp. plantarum and Lactiplantibacillus plantarum subsp. argentoratensis. Since your work is specifically aimed to evaluate a stress response, please clarify this point, a given species (even strain) could behave singly, so it has to be clear if the behaviour is species specific or not.
Response: Thanks for your advice. We have changed Lactobacillus plantarum NMGL2 to Lactiplantibacillus plantarum NMGL2.
- Comment:In general, the abstract should be more descriptive, reporting the main numerical data (percentages, levels, concentrations etc.). As is, the abstract does not stand alone.
Response: Thanks for your advice. We have revised the abstract accordingly.
- Comment: I do not agree with the conclusion “The present study for the first time provides further understanding of the molecular mechanisminvolved in response of L. plantarum to combinational cold and acid stresses in fermented milk, which facilitates potential application of plantarum in fermented foods with enhanced functionality.” Because no speculations on data are provided. In which way your findings are useful? How could it be of help? I am thinking of cold shock response. Fermentation is carried out at mesophilic temperature, so which is the implication of resistance to cold shock?
Response: We have revised it. Changed “The present study for the first time provides further understanding of the molecular mechanism involved in response of L. plantarum to combinational cold and acid stresses in fermented milk, which facilitates potential application of L. plantarum in fermented foods with enhanced functionality” into “The present study for the first time provides further understanding of the proteomic pattern and metabolic changes of Lpb. plantarum in response to combinational cold and acid stresses in fermented milk, which facilitates potential application of Lpb. plantarum in fermented foods with enhanced survivability.” We studied the protein changes in fermented milk during cold storage, not during fermentation, Understanding the proteomic and metabolic changes of Lpb. plantarum during cold storage of fermented milk would facilitate finding ways to improve survivability of the strain during the cold storage.
- Comment: L94-96. If the primers were designed in this work, please provide details on the corresponded positions on a given type strain used for designation. Otherwise, if primers description was published elsewhere, please add reference.
Response:We have added a reference to the revised manuscript providing primer information: Na, L.; Song, M.; Likang, Q. Screening and application of lactic acid bacteria and yeasts with L-lactic acid-producing and antioxidant capacity in traditional fermented rice acid. Food Science & Nutrition 2020, 8, 6095-6111.
- Comment: The strong limit of this work is a control production. Only one strain, the one the authors selected based on in vitro screening, but no comparison with a strain showing the worst characteristic was provided. How can the authors state that the strain chosen behave better than other LAB without a comparison trial? It is as if a control is missing
Response:Thanks for your advice. In Table 1, we have isolated 31 lactic acid bacteria strains, and Lpb. plantarum NMGL2 was screened out based on the assays of tolerance to cold and acid, to be the most tolerant strain among the 31 LAB strains. Therefore, Lpb. plantarum NMGL2 was used for further proteomic study in fermented milk during cold storage.
- Comment: It is not clear the criterion followed to identify bacteria. To do so, bacteria should be isolated following a specific objective, which is not clear from the text, since a main hypothesis is not provided in the abstract. If bacteria resistant to acidic stresses should be collected, then it was better to screen a huge number of bacteria for this character and only those showing interesting characteristic be identified.
Response:Thanks a lot for the comment. The abstract was added at the begining with the introduction sentence: “To understand the mechanism of tolerance of lactic acid bacteria (LAB) during cold storage of fermented milk, 31 LAB strains were isolated from traditional fermented products, and Lactiplantibacillus plantarum NMGL2 was identified with good tolerance to both cold and acid stresses”. So from 31 LAB strains, we focussed on their ability of tolerance to both cold and acid, and strain NMGL2 was selected for further study in fermented milk during cold storage considering the cold and acid stress factors during the storage period.
- Comment: The isolation plan itself is not clear. No number of samples, repetition, characteristics of the samples chosen were provided. Why isolate bacteria from pickle serum to produce a fermented milk? It apparently does not make any sense. Please clarify this very crucial point.
Response:Thanks for your advice. Isolation of LAB strains was performed following a procedure as described in the cited reference. Information about the samples was also provided in the revised manuscript. Lactic acid bacteria are present in both dairy and non-dairy foods, and frequently strains of non-dairy source can also grow well in dairy foods. So samples of both dairy and non-dairy sources were collected for screening in this study, and Lactiplantibacillus plantarum NMGL2 which was isolated from Inner Mongolia cheese with the best tolerance to cold and acid was used in fermented milk for proteomic study.
- Comment: Table 1. This table is not clear too. “Growth of 21 strains of lactic acid bacteria isolated from traditional fermented products in 200 MRS broth at 37°C for 18 h as shown with viable bacterial counts (n = 3, x ± SD).”. In my opinion, the authors should show the levels of viable bacteria in the fermented products used for isolation. This would provide important indications on the possible success of the selected strains in fermented milk. This work does not provide indication on the levels reached in a real food system. Showing the levels reached in synthetic media is not useful, since each bacterium grows alone and is not subjected to the competition with other bacteria. Of course all bacteria grow well in MRS.
Response:Thanks for the comment. MRS medium is commonly used for isolation and evaluation of LAB strains. This medium was also used in this study for initial evaluation of growth characteritics of the strains by measuring the viable counts. So MRS medium was only used for comparing the growth of the strains to select strains with good growth as shown in Table 1, and Lactiplantibacillus plantarum NMGL2 showing good growth in MRS and good tolerance to cold and acid was selected for proteomic study in fermented milk, where strain NMGL2 also showed good growth.
- Comment: L202-203. The results “These strains showed good growth in MRS broth at 37°C for 18 h, with viable counts 202 from 8 log cfu/mL to about 10 log cfu/mL.” do not provide any useful indication. To be eliminated. It would make more sense to show the growth in milk.
Response:Thanks for the comment. As described above, with the commonly used MRS medium the growth characteristic of the strains were initially evaluated, and Lactiplantibacillus plantarum NMGL2 showing good growth in MRS and good tolerance to cold and acid was selected for proteomic study in fermented milk, where strain NMGL2 also showed good growth.
- Comment: Levilactobacillus brevis, not Lactobacillusbrevis (Zheng et al., 2020)
Response:We have modificated it.
- Comment: Lacticaseibacillus casei (Zheng et al., 2020)
Response:We have modificated it.
- Comment: Lactiplantibacillus pentosus (Zheng et al., 2020)
Response:We have modificated it.
- Comment: please revise species name also in Introduction.
Response:We have revised species name in Introduction.
- Comment: Figure 1. Evolutionary tree of L. plantarum NMGL2. First of all it should be reported on which gene. Secondly, I am not sure of its significance to support strain selection.
Response:The gene sequence of the screened Lactobacillus plantarum NMGL2 has been added to the manuscript as an attachment. The evolutionary tree of Lactiplantibacillus plantarum NMGL2 was made for further identification of the strain, and it is not related to tolerance selection of the strain in this study.
Round 2
Reviewer 1 Report
There is still no scientific contribution regarding results presented in Table 1 because optimal growth conditions were applied. Furthermore, the experimental data in tables 2 and 3 are still expressed as absorbance and not as log cfu/ml (as it is in Table 1 and therefore results are not comparable).
Because authors did not accept recommendation about elimination of the results presented in tables 1, 2 and 3 from the text of manuscript because they don’t contribute to the aim of study (only results regarding strain Lactobacillus plantarum NMGL2 are relevant for the aim of study described in the manuscript), the other option is removal of these tables in the supplement material of the manuscript.
Author Response
Q1: There is still no scientific contribution regarding results presented in Table 1 because optimal growth conditions were applied. Furthermore, the experimental data in tables 2 and 3 are still expressed as absorbance and not as log cfu/ml (as it is in Table 1 and therefore results are not comparable).
Response: Thanks for the comment. The reason that we compared the growth of the isolated LAB strains under the optimal condition was just for initial evaluation of the isolated strains in terms of their general performance of growth. Among the 31 isolated strains, 21 of them that showed better growth at 37℃ were further evaluated for growth at low temperatures (4℃, 10℃). From each set of the data in Table 1 and 2, it is observed that strains with better growth at 37℃ (Table 1) also showed better growth at 4℃and 10℃(Table 2). The 5 selected strains showing the best growth at the low temperatures are also the best strains in terms of their growth performance at 37℃, though data in Table 1 and tables 2 and 3 in different units are not comparable.
Q2: Because authors did not accept recommendation about elimination of the results presented in tables 1, 2 and 3 from the text of manuscript because they don’t contribute to the aim of study (only results regarding strain Lactobacillus plantarum NMGL2 are relevant for the aim of study described in the manuscript), the other option is removal of these tables in the supplement material of the manuscript.
Response: Thanks for your advice. We agree that tables 1 is moved to the supplement material of the manuscript, considering the necessary screening process of the isolated strains.
Reviewer 2 Report
The authors provided exhaustive responses to my comments.
However, I am still of the opinion that there is no point in showing growth in MRS, all strains of course are able to grow well and a selection is not possible, I would rather show growth in "skim milk" medium.
Table 2 and throughout the text. Levilactobacillus and Lacticaseibacillus were abbreviated both as L. and this creates confusion. Please revise abbreviation of the different genera
Author Response
The authors provided exhaustive responses to my comments.
However, I am still of the opinion that there is no point in showing growth in MRS, all strains of course are able to grow well and a selection is not possible, I would rather show growth in "skim milk" medium.
Response: Thanks for the comment. MRS medium is commonly used at the stage of isolation and evaluation of LAB strains. In this study, with MRS medium we have successfully isolated and identified the stress tolerant strain of Lactiplantibacillus plantarum NMGL2 in terms of its grow the different temperatures and pHs. To confirm that Lactiplantibacillus plantarum NMGL2 can also grow well in skim milk, more relevant data (Figure 3) about viability of the strain in fermented milk during cold storage are added in the revised manuscript (Line 137-139,263-276).
Table 2 and throughout the text. Levilactobacillus and Lacticaseibacillus were abbreviated both as L. and this creates confusion. Please revise abbreviation of the different genera
Response: we have revised abbreviation of the different genera in this manuscript.